# Established and Emerging Mechanisms in the Pathogenesis of Arrhythmogenic Cardiomyopathy: A Multifaceted Disease

**DOI:** 10.3390/ijms21176320

**Published:** 2020-08-31

**Authors:** Shanshan Gao, Deepa Puthenvedu, Raffaella Lombardi, Suet Nee Chen

**Affiliations:** 1Division of Cardiology, University of Colorado Anschutz Medical Campus, Aurora, CO 80045, USA; Shanshan.gao@cuanschutz.edu (S.G.); Deepa.Puthenvedu@cuanschutz.edu (D.P.); 2Department of Advanced Biomedical Sciences, Federico II University of Naples, 80131 Naples, Italy

**Keywords:** arrhythmogenic cardiomyopathy, pathogenesis, fibroadiposis, molecular pathways, cell-types, inflammation, paracrine factors

## Abstract

Arrhythmogenic cardiomyopathy (ACM) is a heritable myocardial disease that manifests with cardiac arrhythmias, syncope, sudden cardiac death, and heart failure in the advanced stages. The pathological hallmark of ACM is a gradual replacement of the myocardium by fibroadiposis, which typically starts from the epicardium. Molecular genetic studies have identified causal mutations predominantly in genes encoding for desmosomal proteins; however, non-desmosomal causal mutations have also been described, including genes coding for nuclear proteins, cytoskeleton componentsand proteins involved in excitation-contraction coupling. Despite the poor prognosis, currently available treatments can only partially control symptoms and to date there is no effective therapy for ACM. Inhibition of the canonical Wnt/β-catenin pathway and activation of the Hippo and the TGF-β pathways have been implicated in the pathogenesis of ACM. Yet, our understanding of the molecular mechanisms involved in the development of the disease and the cell source of fibroadiposis remains incomplete. Elucidation of the pathogenesis of the disease could facilitate targeted approaches for treatment. In this manuscript we will provide a comprehensive review of the proposed molecular and cellular mechanisms of the pathogenesis of ACM, including the emerging evidence on abnormal calcium homeostasis and inflammatory/autoimmune response. Moreover, we will propose novel hypothesis about the role of epicardial cells and paracrine factors in the development of the phenotype. Finally, we will discuss potential innovative therapeutic approaches based on the growing knowledge in the field.

## 1. Introduction

Arrhythmogenic cardiomyopathy (ACM) is a hereditary disease characterized by fibroadipose infiltration of the myocardium that typically starts from the epicardial layer and mainly affects the right ventricle [1,2,3,4].

Most ACM cases are caused by mutations in genes encoding for proteins of the desmosomes [1,5,6,7,8], multiprotein structures responsible for cell–cell attachment and cytoskeletal linkage to the cell-membrane, but also hubs for pathways regulated at the cell junction [9]. In the heart, desmosomes are part of unique structures connecting cardiomyocytes to one another, known as intercalated disks (IDs) which, in addition to desmosomes, include several other specialized structures: mainly fascia adherens and gap junctions, but also ion channels and transmembrane receptors [10]. Causal mutations in non-desmosomal genes have also been identified in patients with ACM, mainly in genes encoding for components of calcium (Ca^2+^) signaling, nuclear membrane and cytoskeleton [1].

Life threatening arrhythmias and sudden cardiac death (SCD) are the main clinical characteristics of ACM and often manifest in the early stages of the disease, before the occurrence of the histological abnormalities and cardiac dysfunction [3,11]. However, early diagnosis is challenging due to the nonspecific clinical presentation of the disease [3]; for this reason, ACM remains a leading cause of SCD, particularly in the young and athletes [1,3,4,11,12]. Presently there is no causal therapy and the main goals of treatment are to control symptoms, delay the progression toward heart failure, and to prevent SCD with implantable cardioverter defibrillators (ICDs) [1,3,4]. However, ICD implant is associated with significant device-related complications and the selection of candidates for ICD is still unsatisfactory [13,14,15,16]. Therefore, a better understanding of the underpinning mechanisms of early cardiac arrhythmias in ACM could facilitate diagnosis, risk stratification and treatment of ACM patients.

Several molecular and cellular mechanisms have been proposed to explain the pathogenesis of ACM. It has been shown that the presence of the ACM causal mutations induce activation of the Hippo signaling and inhibition of the Wnt signaling and activation of transforming growth factor beta (TGFβ) signaling [1,2,17,18,19,20,21,22,23]. Recent studies suggest electric instability in early stages of ACM may be the consequence of dysregulation of ion channels and Ca^2+^ signaling machinery, not only as direct effect of causal mutations located in genes encoding components of the Ca^2+^ homeostasis, but also as consequence of mutations in desmosomal genes [24,25,26,27,28,29,30,31,32,33]. Moreover, inflammation, a known histological finding of ACM [2,4,34] and autoimmunity are emerging as important players in the development of the phenotype [35,36,37].

The cell origin of fibroadipocytes replacing cardiac myocytes in the context of ACM has been explained by different theories including the trans-differentiation of adult cardiomyocytes [38,39] and the abnormal differentiation of immature cell types [1,18,40,41,42,43]. 

In this review we will focus on the molecular and cellular mechanisms which have been proposed in the pathogenesis of ACM. We will also discuss novel plausible mechanisms such as the possible role of secreted factors from cells carrying the causal mutation and the cross-talk among cardiac cell types which contribute to the development of the ACM phenotype. 

## 2. Genetics, Clinical and Histological Hallmarks

### 2.1. Genetics

ACM is typically transmitted as an autosomal dominant trait with incomplete penetrance and variable expressivity; rarely, the cardiomyopathy is inherited as a recessive trait in the context of the so called cardio-cutaneous syndromes, such as Naxos disease and Carvajal syndrome, in which ACM is associated with palmoplantar keratosis and woolly hair [1,44,45]. About half of ACM cases are caused by mutations in five genes encoding proteins of the desmosomes: namely, plakophilin-2 (*PKP2)* [46,47,48], desmoplakin (*DSP*) [45,49], desmoglein-2 (*DSG2*) [50,51], democollin-2 (*DSC2*) [52,53] and plakoglobin (*JUP)* [44,54]. Among desmosome genes, *PKP2* is the most common causal gene for ACM, accounting for approximately 43% of the cases [5,6,7,8]. Atypical forms of ACM are caused by mutations in non-desmosomal genes such as TGF-beta-3 (*TGFβ3*) [55], cardiac ryanodine receptor (*RYR2*) [56,57], phospholamban (*PLN*) [58,59], transmembrane protein 43 (*TMEM43*) [60], lamin A/C (*LMNA*) [61], desmin (*DES*) [62,63], titin (*TTN*) [64,65], Sodium Voltage-Gated Channel Alpha Subunit 5 (*SCN5A*) [66], N-cadherin (*CDH2*) [67,68], αT-catenin (*CTNNA3*) [69] and filamin C (*FLN*C) [70,71]. A detailed list of the genes whose mutations have been associated with ACM is shown in Table 1. 

### 2.2. Clinical Phenotypic Expressions of ACM

ACM includes different clinical phenotypes; among them, the better defined is the classic arrhythmogenic right ventricular cardiomyopathy (ARVC) which primarily affects the right ventricle [73]. Left-dominant arrhythmogenic cardiomyopathy (LD-ACM) primarily involving the left ventricle and biventricular forms have also been described [74,75]. Furthermore, a growing body of data has shown genetic and clinical overlaps between LD-ACM and a subset of dilated cardiomyopathy (DCM) (so called arrhythmogenic-DCM, a-DCM) which presents with increased arrhythmogenic risk exceeding the degree of the morphological anomalies and systolic dysfunction [74,76,77,78]. On the basis of these observations, an expert panel of the Heart Rhythm Society (HRS) has proposed to include a-DCM, ARVC and LD-ACM in a common nosological entity whose hallmark is the electrical instability [11]. However this classification of ACM has yet to be agreed upon by all the experts in the field. In order to better define ACM, a recently published International Expert Consensus Document has proposed an upgrade of the 2010 Task Force diagnostic criteria [3] for the diagnosis of ACM phenotypic variants [79]; the new proposed diagnostic criteria (so called Padua criteria) include genetics, tissue characterization by cardiac magnetic resonance, depolarization/repolarization ECG abnormalities and ventricular arrhythmia features [79].

A-DCM remains an undefined entity with extensive overlap in genetic background and clinical manifestations with other cardiomyopathies characterized by high risk of SCD. In particular, a clear distinction between LD-ACM and a-DCM remains challenging, especially in the early stages and in cases with mild phenotypic expression. Genetic overlaps and histological characteristics are shown in Figure 1, in panels A and B respectively, while Table 2 summarizes distinctive imaging and electrocardiographic features of ARVC, LD-ACM and a-DCM. 

An interesting genetic overlap has been described between ACM and catecholaminergic polymorphic ventricular tachycardia (CPVT) caused by a mutation in RYR2; however, the role of RYR2 in ACM must still be clarified because of the high background rate of rare missense RYR2 variants [80].

## 3. Molecular Pathogenic Pathways

ACM in most cases is considered a disease of desmosomes, structural proteins with both mechanical and signaling function, the latter mainly involved in cellular proliferation, differentiation and apoptosis, and regulation of gene expression [1,17,18,19,41,81,82,83]. 

Results of in vitro and in vivo studies indicate that abnormalities in biomechanical properties of the desmosome, as well as of the cross-talk between the desmosomes and other cellular structures such as the cytoskeleton, nucleus, gap junctions, and ion channels, contribute to the pathogenesis of ACM [1,17,18,19,41,81,82,83,84]. 

Several studies have identified dysregulation of canonical Wnt, Hippo and TGFβ signaling pathways as responsible for the development of the ACM phenotype, mainly fibroadiposis (Figure 2).

### 3.1. Canonical Wnt Signaling in ACM

The name Wnt is a portmanteau created from names Wingless and Int-1. Wnts are a large family of secreted factors with established roles in embryogenesis and tissue renewal, as they regulate proliferation, polarity, migration and fate determination of progenitor cells [85,86]. Three Wnt signaling pathways have been described: the canonical Wnt, the non-canonical planar cell polarity and the non-canonical Wnt/calcium pathways [87].

The canonical Wnt pathway is β-catenin-dependent and is activated by binding of Wnt ligands to the Frizzled receptor and its co-receptor lipoprotein receptor-related protein (LRP)5/6 [88]. In the absence of Wnt ligands, glycogen synthase kinase-3β (GSK-3β) phosphorylates β-catenin, targeting it to degradation. The interaction between GSK-3β and β-catenin is stabilized by axin, which serves as a negative regulator of the canonical Wnt pathway [89]. Upon binding of Wnt ligands to the receptors, GSK-3β no longer phosphorylates β-catenin, which leads to the stabilization and accumulation of β-catenin in the cytoplasm and its translocation into the nucleus where it binds the TCF transcription factor, triggering downstream signaling cascades [90]. Suppression of the canonical Wnt/β-catenin signaling has been implicated in adipogenesis, fibrogenesis, and apoptosis [91,92,93].

We have demonstrated that inhibition of the canonical Wnt signaling has a critical function in the enhanced adipogenesis in desmosomal ACM [1,17,18,19]. We found that knockdown of DSP in cultured atrial myocytes (HL-1) destabilizes the desmosomes causing the release of (JUP and its translocation to the nucleus. JUP shares structural and functional similarities with β-catenin; hence when it enters the nucleus it competes with β-catenin in the binding to the TCF/LEF1 transcription factor. The final effect of the antagonism between JUP and β-catenin is suppression the canonical Wnt/β-catenin signaling. In this study, DSP-deficient HL1 showed significant upregulation of the adipogenic genes peroxisome proliferator-activated receptor-gamma (PPARγ) and CCAAT enhancer-binding protein–α (C/EBP-α), as well as of the fibrogenesis-related genes *Col1a1*, *Col1a2*, and *Col3a1*. Following incubation in adipogenic differentiation medium, remarkably increased accumulation of fat droplets was observed in DSP-deficient HL1 compared to control cells. Similarly, cardiac-specific deletion of DSP in transgenic mice recapitulated the human ACM phenotype, characterized by ventricular arrhythmias, patchy areas of fibrosis and fat infiltration in heart and cardiac dysfunction [19]. In another study, we showed that mutated JUP, carrying a truncation-mutation, does not incorporate in the desmosomes and accumulates in the nucleus where it competes with β-catenin for binding to the transcription factor TCF/LEF, reducing the transcriptional activity of β-catenin [18]. Moreover, C-kit positive cardiac progenitor cells isolated from the heart of the mutant mice showed increased differentiation to adipocytes through a canonical Wnt signaling suppression mechanism [18]. Suppression of Wnt signaling has been shown in several other ACM models. A mouse model of ARVC subtype 5 expressing a mutated form of the non-desmosomal gene *TMEM43* showed cardiomyocyte death and severe fibrofatty replacement. AKT was found inhibited, leading to GSK-3β activation and consequent inhibition of β-catenin activity in this mouse model [23]. Furthermore, the pathogenic role of inhibition of Wnt signaling in ACM has been shown in an ACM transgenic mouse model with cardiomyocyte-specific overexpression of a FLAG-tagged human desmoglein-2 with the Q558* nonsense mutation [20] and in a zebrafish model of DSP deficiency [21]. Collectively, these results suggest that alteration of Wnt signaling pathway could be an important mechanism underlying the fibrofatty infiltration in both desmosomal and non-desmosomal ACM.

### 3.2. Hippo Signaling in ACM

The Hippo pathway is another evolutionary conserved pathway involved in organ size control and tumor growth [94]. Key components of the Hippo kinase-cascade include STE20-like protein kinase (MST), large tumor suppressor kinases (LATS)1/2), Yes associated protein (YAP) and transcriptional co-activator with PDZ-binding motif (TAZ). YAP and TAZ, which are negatively regulated by the Hippo pathway, function as transcriptional coactivators and when activated, translocate into nucleus where they interact with TEA domain family member (TEAD) transcription factor, leading to transcriptional events which direct cell proliferation, differentiation and apoptosis [95,96].

A previous study from our group showed activation of the Hippo pathway in the hearts of patients with ACM and of two different mouse models of ACM as well as in PKP2 deficient myocytes in culture. In particular, activation of the Hippo signaling and consequent inactivation of YAP by phosphorylation was manifested by increased levels of the upstream Hippo regulator neurofibromin 2 (NF2), of phospho-YAP (pYAP) and phosphoMST (p-MST). The PKP2 deficient cells showed increased adipogenesis as observed in the hearts of ACM mouse models and patients [17]. In addition, activation of the Hippo signaling was responsible for Wnt inactivation through sequestration of β-catenin in the cytosol by pYAP. A recent study on a novel zebrafish model of DSP deficiency has shown that Hippo/YAP-TAZ, Wnt/β-catenin and TGFβ/Smad3 were significantly altered [21]. The findings of these studies strongly indicate a mechanistic association among mutations in desmosomal components, Hippo pathway activation, Wnt inhibition and enhanced adipogenesis in ACM.

### 3.3. TGFβ Signaling in ACM

TGFβ is the major growth factor involved in cardiac fibrosis [97]. Binding of TGFβ family ligands to receptors induces the activation of multiple downstream signaling pathways, eliciting myofibroblast activation and deposition of extracellular matrix [98]. The canonical TGFβ signaling pathway is mediated by Smad proteins while the non-canonical pathway is Smad-independent [99]. A mouse model with cardiomyocyte-specific *Jup* mutation showed cardiac arrhythmia and massive cardiac fibrosis, as well as dramatically increased expressions of TGFβ1 and phospho-Smad2 (P-Smad2) in the heart. Consistently, TGFβ signaling downstream effector TGFBI (TGFβ-induced) was also upregulated in the mutant hearts, suggesting a significant role of canonical TGFβ pathway in the fibrosis formation in ACM [100]. The authors speculated that the pathological mechanical stress induced by desmosome disruption in the mutant heart results in the increased expression of TGFβ1 ligand and thus the activation of TGFβ pathway. 

Furthermore, it has been shown that levels of pro-inflammatory cytokine interleukin-1β (IL-1β) are elevated in ACM patients [101]. IL-1-induced pathways are known to lead to activation of JNK, a crucial component of the non-canonical TGFβ pathway [102,103]. Hence, in addition to the canonical pathway, the non-canonical TGFβ pathway may potentially contribute to increased fibrosis in ACM as well, but more solid scientific evidence needs to be generated in order to confirm this hypothesis. 

## 4. Electrical Instability in ACM

A distinct feature of ACM compared to other forms of cardiomyopathy is the early occurrence of arrhythmias, before the development of the histological abnormalities and cardiac dysfunction [25]. The majority of the desmosome gene mutations in ACM are loss of function mutations which cause a significant reduction of the expression levels of desmosomal protein at the IDs of the cardiac myocytes. The consequent IDs’ remodeling may impair the electro-mechanical connections between cells. Several studies have shown that structural remodeling of the IDs in ACM hearts causes a reduction in the protein levels and dysfunction of the sodium channel and of connexin 43 (Cx43) [24,26,27,28]. Mutations in non-desmosomal genes such as *PLN* and *RYR2* directly impair the calcium machinery; moreover, deficiency of the desmosomal protein PKP2 has also been associated with abnormal calcium handling. In summary, there is evidence that both desmosomal and non-desmosomal gene mutations cause electro-mechanical dysfunction resulting in increased risk of arrhythmias in ACM patients, independently from the histological abnormalities. Figure 3 synthesizes the proposed mechanisms of cell membrane electrical instability and abnormal Ca^2+^ homeostasis in ACM.

### 4.1. Overview of Excitation-Contraction Coupling in Heart and its Dysfunction in ACM

Cardiac myocytes are specialized contractile cells in which the contraction is the end result of a multistep biological process known as excitation-contraction (E-C) coupling which converts electrical stimulus (excitation) into mechanical force (contraction). The E-C coupling process is initiated by the electrical excitation of the membrane (action potential) which activates the voltage-gated sodium channel (Nav1.5) and triggers the influx of sodium (Na^+^) which in turn induces Ca^2+^ influx through the voltage-sensitive L-type calcium channels (LTCCs). The increase in intracellular Ca^2+^ concentration activates the RYR2 and triggers a flash release of Ca^2+^ from the sarcoplasmic reticulum (SR), a process known as calcium-induced calcium release. The abrupt release of Ca^2+^ from the SR activates the cross-bridge cycling of the myofibrils resulting in shortening of the sarcomere and cardiac muscle contraction (Figure 3) [12]. After contraction, the cardiac myocytes rapidly go into the relaxation phase, in which the cross-bridge cycling is interrupted because of reduction of the levels of Ca^2+^ which is sequestered into the SR by the SR Ca^2+^-ATPase type-2a (SERCA2a) and extruded from the cell by the Na^+^/Ca^2+^-exchanger type-1 (NCX1). The reuptake of Ca^2+^ by SERCA2a into the SR is enabled upon phosphorylation of phospholamban (PLN), an inhibitor of SERCA2a. 

As the desmosomes are localized on the cell membrane and the first step of the E-C coupling is the electrical excitation of the membrane, a dysfunction of the desmosomes, which is typical of ACM is expected to affect action potential generation and propagation. Additionally, ACM mutations have been associated with dysregulation of the intracellular calcium machinery. Hence, all the phases of the E-C coupling are affected in ACM and this contributes to the electrical instability and consequently the high risk of life-threatening arrhythmias, and eventually to a reduced contractility of the cardiomyocytes.

### 4.2. Effects of Desmosome Mutations on the Electrical Properties of the Cell Membrane

Electrical instability in desmosomal ACM may be due to the dysfunction and/or mis-localization of components of the gap junctions and/or ion channels because of the negative effects of the desmosome protein mutations on the ID organizzation.

In vitro and in vivo studies have shown that disruption of the mechanical cell junction machinery due to a desmosome protein mutation is associated with abnormalities in the gap junction machinery. In particular, decreased expression of PKP2 in cardiomyocytes disrupts the normal localization and conductivity of the gap junction protein Cx 43 [29]. Likewise, studies in a cardiomyocyte-specific Dsp-deficient mouse model have shown that loss of DSP affects expression levels, phosphorylation and function of both Cx 40 andCx 43, independently of the cell–cell detachment and prior to the fibro-fatty infiltration of the myocardium [30]. The molecular effects on the connexin system cause right ventricular conduction defects and ventricular arrhythmias that are exacerbated with exercise and catecholamine stimulation [30]. These data suggest a role for PKP2 and DSP as stabilizers of the gap junction integrity and highlight the molecular mechanisms of early electrical defects found in ACM patients. 

Additionally, mutations in desmosomal proteins have been shown to affect the sodium current prior to cardiomyopathic changes [28,32] Other molecules, not directly involved in intercellular coupling, are known to reside at the IDs; among them is NaV1.5, the major α subunit of the cardiac sodium channel [104]; therefore it is plausible that desmosome proteins interact with components of the sodium channel. Indeed, it has been shown that the desmosomal protein PKP2 and NaV1.5 are part of the same molecular complex and that loss of PKP2 affects the amplitude and kinetics of the sodium current, and the action potential propagation in a monolayer of cardiomyocytes [27].

### 4.3. Ca^2+^ Homeostasis and Non-Desmosomal ACM Causal Genes

Among the non-desmosomal ACM causal genes, *RYR2* and *PLN* encode for proteins implicated in Ca^2+^ cycling. 

Mutations in genes encoding Ca^2+^ channels or regulators of Ca^2+^ homeostasis have been shown to cause inherited arrhythmogenic diseases such as catecholaminergic polymorphic ventricular tachycardia (CPVT), Brugada syndrome, Timothy syndrome, and long and short QT syndromes [56,59,105]. Therefore, it is not surprising that mutations in Ca^2+^ cycling genes could be implicated in cardiomyopathies at elevated arrhythmic risk such as a-DCM and ACM. We will focus on *PLN* and *RYR2* as they are well-known causal genes for a-DCM and ACM.

PLN is a regulator of the activity of SERCA, the pump which actively removes Ca^2+^ from the cytoplasm back into the SR after each contraction. During contraction, dephosphorylated PLN interacts with SERCA and inhibits its activity, while during cardiac relaxation, phosphorylation of PLN reduces its inhibitory effect on SERCA. Mutations in *PLN* are known to cause a-DCM and ACM and among the identified mutations, the PLN-R14Del mutation is the most prevalent cardiomyopathy-associated mutation in the Netherlands (founder effects) [59,105].

The PLN-R14Del mutation causes the deletion of the highly conserved basic amino acid Arg-14, which induces the loss of the ability of PLN to relieve from SERCA [106,107,108]. Thus far, approximately 2000 carriers with the PLN-R14Del mutation have been identified [59,109]. The clinical phenotype is extremely variable, ranging from totally asymptomatic subjects to patients with severe dilated cardiomyopathy and elevated risk of life-threatening arrhythmias. The observed variability of the phenotype raises the possibility that a yet to be identified pathogenic modifier variant could be present with the PLN-R14Del mutation and contribute to the development of the cardiomyopathy. Histological analysis of cardiac tissue in PLN-R14Δ/Δ mice at 8 weeks of age showed significant increase in cardiac fibrosis but it is not known if mutations in PLN could induce adipogenesis in this mouse model [110].

Ryanodine receptor 2 (*RYR2*) has been the first causal gene identified in ACM patients in an autosomal dominant form, namely ARVD2 (OMIM 600996) [111]. The described clinical presentation of ACM due to *RYR2* mutations is characterized by polymorphic and effort-induced ventricular arrhythmias and high risk of sudden cardiac death. However, the role of *RYR2* as a causal gene for ACM is still debated as there are insufficient functional data to prove the correlation with fibroadiposis which is the typical histological characteristic of ACM.

*RYR2* encodes one of the components of a large calcium channel localized on the SR membrane which is responsible for the release of Ca^2+^ from the SR into the cytoplasm during the E-C process in cardiomyocytes (Figure 2). The RYR2 channel is composed of a tetramer of the ryanodine receptor and a tetramer of FK506 binding protein 12.6 (FKBP12.6); the latter stabilizes the closed state of the channel. Thus far, four missense mutations (R176Q, L433P, N2386I and T2504M) in the *RYR2* gene have been identified in families with ACM. All these mutations are localized in the regulatory domain of RYR2 and impair its interaction with FKBP12.6 in stabilizing the tetrameric structure of the channel [56]. As a result, mutant RYR2 proteins fail to retain Ca^2+^ in the SR-enhancing spontaneous release of Ca^2+^ which leads to delayed after-depolarizations (DADs) and ventricular arrhythmia in ACM patients [56]. This phenomenon has been proven in a transgenic mouse model carrying the R176Q mutation which presents with ventricular tachycardia due to spontaneous SR Ca^2+^ release as well as in both paced and non-paced isolated cardiomyocytes [112]. However, the histologic analysis of the *RyR2^R176Q/+^* mice hearts did not detect infiltration of fibro-adipocytes or structural abnormalities typical of ACM [112].

### 4.4. Ca^2+^ Homeostasis and Desmosomal ACM Causal Genes

Recently, it has been reported that desmosomal mutations are associated with aberrant Ca^2+^ homeostasis.

A dominant in-frame insertional mutation in JUP has been shown to create a novel interaction site between mutant plakoglobin and histidine-rich calcium-binding protein (HRC-BP) in an in vitro yeast-two-hybrid screening [54]. HRC-BP is a high capacity Ca^2+^ binding protein localized in the SR of cardiac and skeletal muscle cells where it controls the amount of Ca^2+^ to be released from the SR [113]. It is possible that the interaction of the mutant JUP with HRC-BP could affect calcium homeostasis and thereby promote myocyte injury and contribute to electrical instability in ACM. 

More recently, several studies in different in vitro and in vivo modes, have shown that loss of PKP2, the most frequent ACM causal gene, is associated with alteration of Ca^2+^ machinery. Ca^2+^ handling dysregulation has been reported in human-induced pluripotent stem-cell-derived cardiomyocytes (hIPSC-CMs) deficient in PKP2 which exhibited decreased expression levels of SERCA and slower intracellular calcium reduction during relaxation [114]. Studies in a cardiac-specific PKP2-deficient mouse model have shown that PKP2 is necessary for the transcription of genes that control intracellular calcium cycling. In particular, lack of PKP2 reduced expression of *Ryr2*, *Ank2Cacna1c* (coding for CaV1.2) and *Trdn* (coding for triadin), and protein levels of calsequestrin-2 (CASQ2). These molecular alterations caused disruption of intracellular calcium homeostasis and increased the susceptibility to develop isoproterenol-induced arrhythmias in the mouse model [31]. 

Later on, a combination of imaging, biochemical and high-resolution mass spectrometry was used to assess functional/structural properties of cells/tissues derived from cardiomyocyte-specific, tamoxifen-activated, PKP2 knockout mice (PKP2cKO) in which PKP2 deletion was induced at a time point preceding an overt electrical and/or structural phenotype [33]. In this paper the authors studied cardiomyocytes from right (RV) or left ventricular (LV) free walls separately and found that while PKP2cKO cardiomyocytes from the LV were not different from wild type controls, PKP2cKO cardiomyocytes from the RV showed several alterations in Ca^2+^ handling. In particular they presented increased amplitude and duration of Ca^2+^ transients, increased levels of Ca^2+^ in the cytoplasm and in the SR, and increased frequency of spontaneous Ca^2+^ release events (sparks) [33]. The authors speculated that in the absence of PKP2, increased membrane permeability of the cardiomyocytes enabled the Cx43 to act as the conduits for Ca^2+^ entry in the cytoplasm. In addition, PKP2cKO-RV cardiomyocytes presented enhanced Ca^2+^ sensitivity of the RYR2 and Ca^2+^ accumulation in the mitochondria. These molecular changes were associated with the occurrence of early- and delayed- aftertransients in RV myocytes and increased susceptibility to arrhythmias in Langendorff-perfused hearts from the transgenic mice [33]. Furthermore, genetic ablation or pharmacologic inhibition of Cx43 normalized Ca^2+^ homeostasis, while inhibition of protein kinase C normalized spark frequency [33]. Therefore, the authors speculate that the LV-RV asymmetric Ca^2+^ dysregulation triggers early arrhythmias and induces myocyte damage which initiates the structural changes observed at a later stage of ACM.

## 5. Cell Sources of Fibro-Adipocytes in ACM

Several hypotheses have been proposed to explain the replacement of cardiac myocytes with adipocytes in ACM, from the trans-differentiation of adult cardiomyocytes to the abnormal differentiation of cardiac stem cells. Despite the considerable progress made in this field, the cell source of fibrosis and adipocytes in ACM remains an enigma.

### 5.1. Trans-Differentiation of Adult Cardiomyocytes

The hypothesis that cardiomyocytes trans-differentiate into adipocytes in ACM hearts was proposed, based on a case study, as an alternative hypothesis to the dystrophic theory, which considered ACM a defect of development present at birth [38]. In this case report, the author found that myocytes adjacent to adipose tissue showed multiple sarcoplasmic vacuoles and were positive for desmin and vimentin. Hence, the author concluded that the vacuolated cells were preadipocytes trans-differentiated from myocytes. The data were not conclusive because, although ultrastructural examination showed the lipid nature of the sarcoplasmic vacuoles, vimentin is not a specific marker for preadipocytes but is a specific marker for fibroblasts instead. Another case report in ACM published by a Japanese research group showed accumulation in cardiomyocytes of a large number of intracellular lipid droplets of varying sizes which were released in the interstitial space of the myocardium through disruptions of the plasma membrane [39]. A recent study has shown that cardiomyocytes can be de-differentiated by upregulating signaling pathways implicated in cell cycle proliferation, survival and metabolism [115]. However, how the causal mutation of ACM may induce the trans-differentiation of adult cardiomyocytes to adipocytes has not been proven. Therefore, the cardiomyocyte trans-differentiation hypothesis in ACM hearts remains debatable due to the lack of reproducible immunohistological and molecular evidence.

### 5.2. Cardiac Progenitor Cells (CPCs) switch from Myogenesis to Adipogenesis

Cardiac progenitor cells ^+^) expressing desmosomal proteins have been postulated as alternative cell sources of adipocyte in ACM hearts [18,43]. These cells are considered a more plausible candidate, because, although of cardiac lineage, they are immature and hence more prone to deviations in their differentiation path.

Through genetic fate-mapping experiments, Lombardi et al. have shown that cardiac progenitors from the second heart field differentiate to adipocytes in the heart of ACM mice with cardiac-specific deletion of desmoplakin (Dsp) [43]. The authors also corroborated the finding in human hearts with ACM. Notably, the authors identify inhibition of the canonical Wnt signaling, which is known to regulate myogenesis and adipogenesis [116,117]., as the main mechanism of increased adipogenesis in their mouse model [43] The study provides suggestive evidence that progenitor cells from the second heart field in the presence of desmosomal mutation deviate from their myogenic fate to an adipogenic fate upon suppression of canonical Wnt signaling. 

The same group generated transgenic mice overexpressing either wild-type or truncated plakoglobin (JUP) [18], another known ACM causal gene. The hearts of transgenic mice expressing truncated JUP showed increased adipogenesis and fibrosis. The authors isolated ckit+/Sca1+ cells from the heart of the transgenic mice, and showed increased adipogenic differentiation of these cells due to suppression of Wnt signaling caused by nuclear translocation of truncated JUP. Nonetheless, c-Kit+ cardiac progenitor cells in adult hearts are rare and account only for a small fraction of adipocytes in ACM, suggesting that other cell types may be involved in the development of fibroadiposis [40,42].

### 5.3. Mesenchymal Cells and Cardiac Fibro-Adipocyte Progenitors Differentiation to Adipocytes

Two groups have shown that cardiac resident cells of mesenchymal origin express desmosome proteins and are a source of adipocytes in ACM [40,42]. These findings for the first time implicate a non-cardiogenic cell type in the pathogenesis of ACM, previously considered a disease exclusively of the cardiomyocytes.

Previous studies in the skeletal muscle of patients with Duchenne muscular dystrophy (a disease that also shows pathological replacement of muscle with fibroadiposis as in ACM) described a resident cell population identified by the surface marker platelet-derived growth factor receptor-α (PDGFRα) as a major contributor to abnormal tissue repair. These cells were named fibro-adipocyte progenitors (FAPs) because of their ability to differentiate into both fibroblasts and adipocytes [118,119]. Since ACM shows fibroadipocytic replacement of the myocardium, Lombardi et al. hypothesized that the heart may contain similar resident FAPs [42]. Indeed, the group isolated a specific cardiac cell population from human and mouse hearts by fluorescence-activated cell sorting (FACS), using a set of cell surface markers in order to select cells expressing PDGFRA and exclude those expressing established lineage (CD32, CD11B, CD45, Lys76, Ly−6c and Ly6c) and fibroblast markers (thymocyte differentiation antigen 1, and discoidin domain receptor 2) [42]. The authors show that cardiac-FAPs express desmosomal proteins and are bipotential, as the majority of them express the fibroblast marker collagen 1 α-1, while a small subset express the adipogenic marker CEBP-α [42]. Interestingly, the desmosome protein desmoplakin was predominantly expressed in the adipogenic but not in the fibrogenic subset of cardiac FAPs [42]. The same group, by in vivo genetic fate-mapping experiments, showed that 40% of adipocytes in the heart of a mouse model of ACM originate from FAPs through a Wnt-dependent mechanism [42]. 

Cardiac mesenchymal cells (C-MSC) described in the paper of Sommariva et al. [40], and cardiac fibroadipocytes (FAPs) described by Lombardi et al. [1], were isolated by two different methods and by using different cell surface markers; hence, the C-MSC and the cardiac FAPs described by the two groups cannot be considered exactly the same cell population. Despite the differences, a significant overlap between these two cell populations is expected, as FAPs are most likely a subpopulation of the C-MSC cell-pool.

### 5.4. Epicardial Progenitor Cells (EPCs)

Epicardial progenitor cells (EPCs) are known to play an important role in heart development by two main mechanisms: 1) differentiation into several cell types through epithelial to mesenchymal transition (EMT) and 2) secretion of several molecules which regulate the fate of neighboring cells [120]. In adult hearts, the regenerative capacity of the epicardium is significantly reduced, but EPCs may be reactivated by acute injury, such as after myocardial infarction [120,121,122]. However, adult EPCs undergo limited EMT; this suggests that their contribution to cardiac repair may be exerted predominantly through paracrine mechanisms [123,124]. It has been shown that the Hippo signaling, which has been implicated in the pathogenesis of ACM [17], is activated by mechanical stress and has a pivotal role in proliferation and differentiation of EPCs during cardiac organogenesis [125,126]. Furthermore, it has been shown that epicardial Hippo signaling plays a key role in adaptive immune regulation during the post-MI recovery phase [127]. 

In ACM, fibroadiposis typically starts from the epicardium, hence EPCs are plausible regulators of the abnormal tissue homeostasis, mainly at the onset of the disease. We and others have shown that EPCs express desmosomal proteins [128,129]. On the basis of this observation, we can hypothesize that the desmosomal mutation may cause chronic mechanical injury in EPCs due to cell–cell detachment and induce their activation; as a result, mutant-activated EPCs may directly differentiate to fibroadipocytes and/or secrete paracrine factors affecting the differentiation of neighboring cells. 

EPCs have been implicated as a cell source for adipocytes in the heart in different cardiac diseases [130,131]. Yamaguchi et al. showed that epicardial cells are able to convert into adipocytes upon activation of peroxisome proliferation activator receptor gamma (PPARG) in the heart of mice [131]. 

In ACM, epicardial cells have been proposed as a source of adipocytes: Matthes et al. [128] showed that siRNA-mediated knockdown of Plakophilin-2 in epicardial cells isolated from neonatal rat hearts increases proliferative rate and migration velocity and induces lipid accumulation in these cells. However, the experiments of this study have been conducted exclusively in vitro and must be confirmed in in vivo models. Preliminary in vivo studies [129] have shown that DSP deletion in EPCs, identified by the Wilms Tumor 1 (WT1) marker, induce fibroadiposis in the heart, typically starting from the epicardial layer. Genetic fate mapping experiments in this novel mouse model showed that only a small fraction of fibroadipocytes derive directly from mutant EPCs, suggesting a main paracrine role of the epicardium in ACM. The effects of ACM desmosomal mutations in epicardial cells remain a subject for future investigation.

## 6. Novel Mechanisms in the Pathogenesis of ACM

### 6.1. Inflammation and Autoimmunity in ACM

Inflammation is a feature of early stages of ACM, detected before the deposition of fibrofatty tissue, as shown by studies in both patients and mouse models [34,132]; however, its role has never been studied in detail. Inflammation in the context of ACM may be triggered by cell–cell detachment and activate chemokine- and leukocyte-driven responses resulting in abnormal tissue repair and consequent fibradiposis.

Inflammatory cellular infiltrates have been detected in the heart in over 70% of ARVC patients and in ACM mouse models [2,4,22,34,132]. Inflammation in ARVC patients has also been confirmed by detection of increased plasma levels of IL-1β, IL-6, and TNF-α [133]. Currently, it remains to be determined if inflammation is a primary cause or is a secondary effect due to myocyte death in ACM.

To define the primary role of inflammation in the pathogenesis of ACM, a recent paper has characterized the role of nuclear factor-κB (NFκB), a known master regulator of cellular inflammatory responses in in vitro and in vivo ACM models and in cardiac myocytes from patient-induced pluripotent stem cells [37]. The authors of this paper show that NFκB is activated in cardiac myocytes in ACM and induces the expression of large amounts of inflammatory cytokines and chemotactic molecules [37]. Furthermore, they show that Bay 11–7082, a small molecule inhibitor of NFκB signaling, prevents the development of ACM disease features [37]. The findings suggest that inflammatory signaling directly drives key features of the disease and that targeting inflammatory pathways may be an effective new mechanism-based therapy for ACM.

The role of autoimmunity as a trigger for immune response is a novel research field in ACM. Autoimmunity has been speculated to be the cause of the noninfectious myocarditis in ACM. Specific serum anti-desmoglein-2 auto-antibodies have been detected in both adult and childhood ARVC patients regardless of the underlying mutation [35]. This finding has been also confirmed within a validation cohort and in a Boxer dog model of ARVC [36]. Recently, anti-heart autoantibodies (AHA) and anti-intercalated disk autoantibodies (AIDA) have been detected in the serum of the majority of familial and sporadic ARVC cases, including some healthy relatives [35]. Furthermore, the authors of this study showed that the presence of AHA and AIDA in serum was associated with the severity of the disease [35]. The findings provide suggestive evidence that autoimmunity could play a major role in the pathogenesis of ACM, as previously observed in primary dilated cardiomyopathy [134]. Although these studies facilitated the potential therapeutic use of immunosuppression in biopsy-proven virus-negative autoantibody-positive inflammatory ACM, the exact trigger and mechanisms for the autoimmunity in ACM is not known. A plausible hypothesis could be that the causal mutations may unmask “cryptic” epitopes and trigger the activation of autoimmunity. Moreover, an alternative hypothesis could be that the release of auto-antigens by the damaged myocytes may stimulate the immune system to generate autoantibodies. Longitudinal studies are necessary to clarify whether AHA and AIDA may be used as biomarkers to predict the development and severity of the disease in ACM patients and healthy relatives.

### 6.2. Paracrine Factors as Regulators of Fibroadiposis in ACM

It is well established that cardiac resident cells, including adult cardiomyocytes, are able to release numerous molecules which exert paracrine effects on the surrounding interstitial milieus. 

It has been shown that cardiac progenitor cells (CPCs) expressing the stem cell marker c-kit are present in the adult human heart [135] and that their injection into the injured heart improves cardiac function mainly through the release of paracrine factors [136,137]. Lombardi et al. have shown that ckit+ CPCs express the desmosome protein JUP and are a source of adipocytes in the heart of a mouse model of ACM [18]. However, the adipogenic rates of these cells is too low [40], suggesting that they may contribute to lipid accumulation in ACM mainly though a paracrine function rather than by direct differentiation to fibroadipocytes. 

The epicardium has a well-known paracrine function in physiologic and pathologic conditions [120,123,124,127]. However, its role in the pathogenesis of ACM has never been investigated. Preliminary data from our group in a novel mouse model with deletion of DSP specifically in EPCs and in cultured EPCs [129] raise the intriguing possibility of paracrine mechanisms in the pathogenesis of ACM. Additional studies are ongoing to prove the paracrine role of the epicardium in ACM.

### 6.3. Effects of ACM Causal Mutations on Different Cell Types

Taken together, the studies performed in cell culture and animal models of ACM suggest that the pathogenesis of the disease is complex as it is the result of the combination of diverse effects of the causal mutation on multiple pathways and biological processes which in turn may have different implications in different cell types. Activation of Hippo signaling and TGF-β and suppression of canonical Wnt signaling have been proven in different ACM in vivo and in vitro models and in ACM patients [17,18,19]. However, the effects of these pathways in different cell types have not been studied in detail. It has been shown that desmosomal proteins are not exclusively expressed in cardiomyocytes but also in other cardiac cell types such as CPCs, cardiac FAPs and EPCs [1,18,40,42,43,128,129].

Each of these cell types may respond differently to the presence of the ACM causal mutation (Figure 4): the effects on adult cardiomyocytes may be mainly related to the loss of mechanical continuity and proper ion channel localization and function while mutations in desmosomal genes in CPCs, FAPs and EPCs may affect their migration, proliferation, and differentiation and/or their paracrine function. In addition, cells that are not known yet to express desmosomes may also be affected by factors secreted by the mutant cells. Among them, possible candidates are inflammatory cells which may be induced to migrate and infiltrate the myocardium by chemotactic factors secreted by the mutant cells.

### 6.4. Cell Types Cross-Talk in ACM

The heart is composed of diverse cell types, including several populations of endogenous cardiac progenitor cells. Furthermore, multiple signaling pathways play essential roles in cardiac tissue homeostasis. Different cardiac cell types may respond to the desmosome mutation not only by changing their own cell behavior but also by affecting the neighboring cells through different mechanisms which include: (1) direct contact; (2) release of soluble factors which may act in autocrine, paracrine and endocrine ways [138]; (3) release of complex structures such as extracellular vesicles and exosomes [137,138,139,140]. Each one of these mechanisms may be affected by the presence of the ACM causal mutation.

On the basis of these considerations, ACM must be considered a multifaceted disease whose phenotypic effects are the result of complex cellular and molecular interactions as shown in Figure 4.

## 7. Novel Therapeutic Options in ACM

The main objectives of clinical management of ACM patients are to reduce sudden cardiac death caused by arrhythmias and to prevent disease progression to heart failure. Implantable cardioverter defibrillators (ICD) remain the core therapy to prevent sudden cardiac death and represent the first treatment approach for high-risk patients. Current therapeutic and preventive management options for ACM patients are palliative but not curative. Hence, novel effective therapeutic options are needed.

Molecules able to activate the Wnt signaling or inhibit the Hippo pathway could be therapeutic options; however these molecular pathways are expressed all over the body and their manipulation may have severe collateral effects. Unfortunately, it is challenging to obtain cardiac-specific compounds.

The recent findings supporting the involvement of Ca^2+^ signaling in the pathogenesis of ACM place the foundation for future studies in patients with ACM to examine the role of intracellular calcium homeostasis as a therapeutic target. For instance, a number of studies indicate that treatment with flecainide, a drug which is known to suppress arrhythmias in catecholaminergic polymorphic ventricular tachycardia (CPVT) caused by mutations in the RyR2, is also effective in suppressing arrhythmias in animal models and patients with ACM [31,33,141]. These findings have provided the rationale for an ongoing pilot randomized clinical trial to determine whether flecainide can reduce ventricular arrhythmias in high-risk ACM patients with implantable cardioverter-defibrillators (ICD) (ClinicalTrials.gov Identifier: NCT03685149).

The discovery that cardiac myocytes express inflammatory cytokines and chemotactic molecules in ACM and that pharmacological inhibition of NF-κB signaling prevents the development of the phenotype [37] suggests that anti-inflammatory drug therapy may be an effective, mechanism-based strategy to reduce myocardial damage and risk of SCD in ACM patients. Moreover, the identification of an autoimmune response in ACM patients and mutation carriers and its correlation with disease severity [35] further supports anti-inflammatory drugs as an optimal therapeutic option in ACM.

The upcoming studies on paracrine mechanisms in the onset and development of the disease are expected to have enormous translational implications as they may lead to the identification of novel diagnostic and prognostic biomarkers and therapeutic targets. Moreover, given its accessibility, the epicardium could be a future target for gene therapy or drug-delivery in patients with ACM.

Further discovery of the molecular and cellular mechanisms are necessary to identify more effective therapeutic targets and provide a definitive cure to ACM. Moreover, larger numbers of patients and longer follow-ups are needed in prospective studies/registries and collaborative multi-center randomized controlled trials are required to obtain population-based data in order to provide evidence-based clinical management to ACM patients.

## 8. Conclusions

This manuscript provides a comprehensive review of the established molecular and cellular mechanisms as well as novel hypotheses on the pathogenesis of ACM. Despite the great amount of data accumulated in last 10 years, our knowledge about the responsible molecular pathways and cell-type specific effects of the causal mutations remains unsatisfactory as no effective treatment is currently available for ACM patients. In order to improve ACM patient clinical management, further efforts are needed to expand our understanding of the complex interplay among the biological processes contributing to the development of the phenotype.

## Figures and Tables

**Figure 1 ijms-21-06320-f001:**
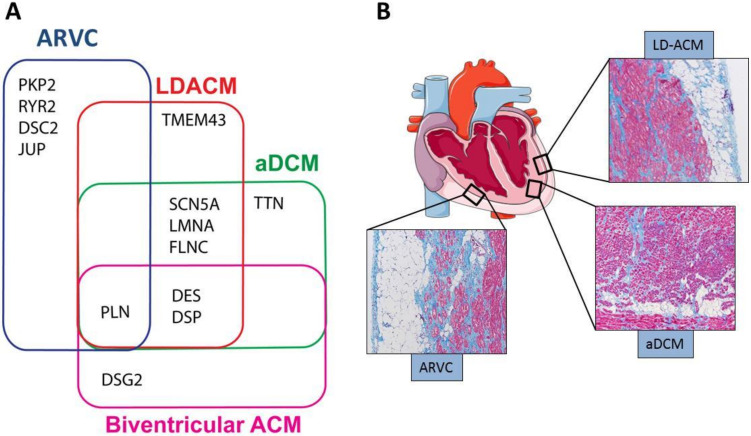
Genetic and histological hallmarks of ACM (arrhythmogenic cardiomyopathy). Panel (**A**) shows the genetic overlaps among the main ACM clinical phenotypes and a-DCM (arrhythmogenic dilated cardiomyopathy). Panel (**B**) shows the typical histological characteristics of ACM: fibroadiposis involving the epicardial layer of the RV (right ventricle) in ARVC (arrhythmogenic right ventricular cardiomyopathy), the epicardial area of the LV (left ventricle) in LD-ACM (left-dominant arrhythmogenic cardiomyopathy) and the midwall of the LV in a-DCM.

**Figure 2 ijms-21-06320-f002:**
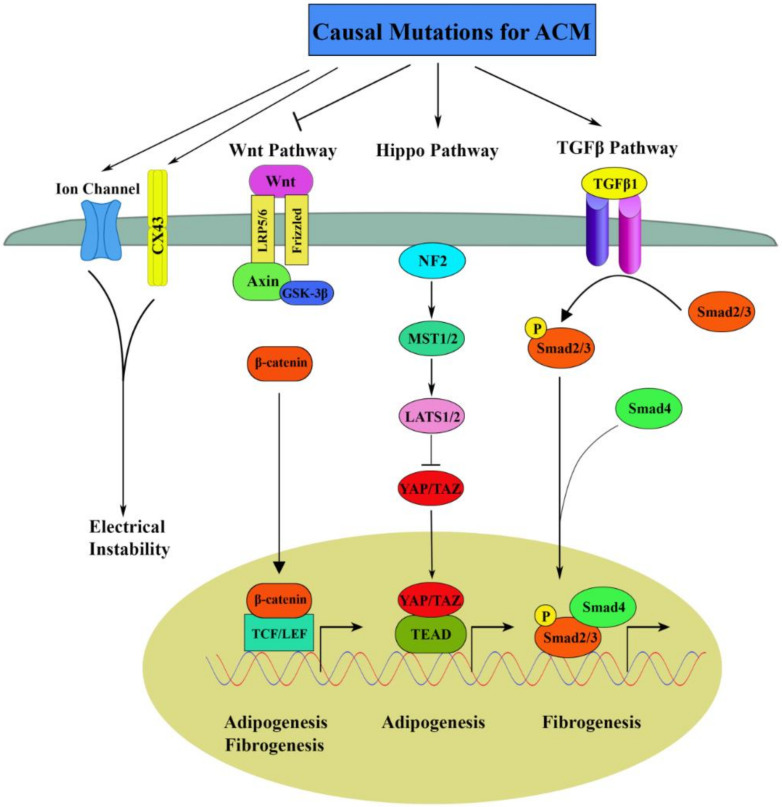
Molecular pathogenic signaling pathways in ACM. Mutations in genes encoding desmosomal components cause Hippo pathway activation, Wnt inhibition and enhanced adipogenesis in ACM. Moreover, the canonical TGFβ pathway contributes to increased fibrosis in ACM through activation of SMADs and JNK, respectively. Furthermore, mutations in desmosome genes destabilize the intercalated disks causing dysfunction of ion channels with consequent electrical instability, a prominent phenotype in the early stages of ACM. ACM: arrhythmogenic cardiomyopathy; Cx43: connexin 43; Wnt: wingless-Int; LRP)5/6: lipoprotein receptor-related protein; GSK-3β: glycogen synthase kinase-3β; TCF/LEF: T-cell factor/lymphoid enhancing factor; NF2: neurofibromin 2; MST1/2: mammalian Sterile 20-related 1 and 2; LATS1/2: arge tumor suppressor 1 and 2; YAP: Yes associated protein; TAZ: transcriptional co-activator with PDZ-binding motif; TAED: TEA domain; TGF-β: transforming growth factor-beta; Smad: small mother against decapentaplegic.

**Figure 3 ijms-21-06320-f003:**
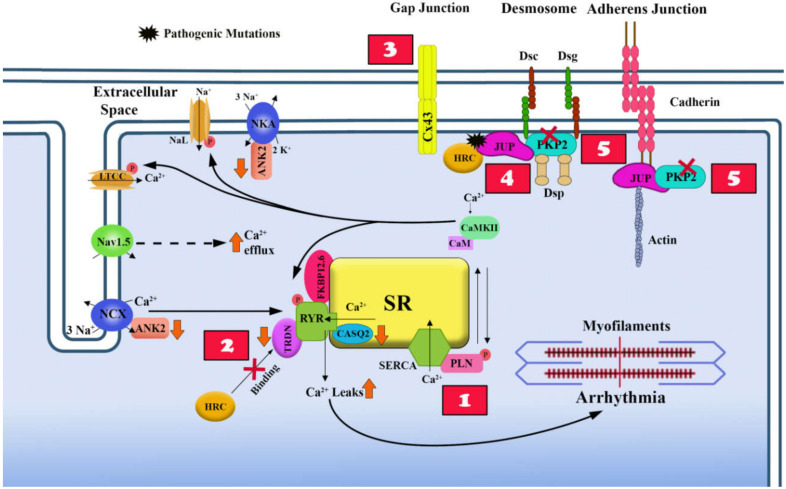
Proposed mechanisms causing arrhythmias in ACM. In healthy myocytes, the voltage-gated sodium channel (Nav1.5), the sensitive L-type calcium channels (LTCCs), the ryanodin receptors (RYR2), and SERCA are appropriately positioned to regulate the calcium induced calcium release cycle. Likewise, the connexin 43 (Cx43), plakoglobin (JUP), plakophilin 2 (PKP2) are positioned at desmosomes and adheren junctions to form a solid intracellular connection to the heart and to regulate membrane trafficking of small molecules as well as electrical signals for coordinated contraction of the heart. In arrhythmogenic cardiomyopathy (ACM), normal calcium signaling in the heart seems to be perturbed. (1) Mutations in *PLN* prevent its ability to relieve SERCA, which repress the re-uptake of cytoplasmic Ca2+ back into the sarcoplasmic reticulum. (2) Mutations in RYR2 impair its interaction with FK506 binding protein 12.6 (FKBP12.6) in stabilizing the tetrameric structure of the channel and enhances Ca2+ leak into the cytoplasm. (3) Cx43-induced arrhythmias in the context of ACM are due to its mis-localization at the membrane secondary to structural defects caused by mutant desmosomal proteins. (4) Mutant JUP sequesters histidine-rich calcium-binding protein (HRC-BP) causing calcium leak into the cytoplasm. (5) PKP2 deficiency decreases levels of Ryr2, Ankyrin-B (Ank2), Triadin (TRDN) and Calsequetrin-2 (CASQ2), contributing to intracellular calcium homeostasis disruption.

**Figure 4 ijms-21-06320-f004:**
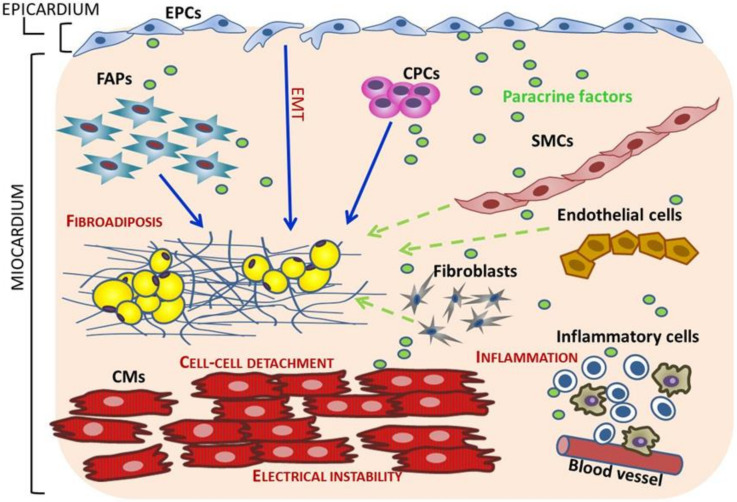
Cell type contributions to the pathological hallmarks of ACM. In cardiomyocytesthe main effects of the presence of the ACM mutation are cell–cell detachment and electric instability due to disorganization of the intercalated disks. Fibroadiposis may be due to the direct differentiations of cells expressing the mutation or to the abnormal cell-behavior induced by secreted factors from mutant cells. Direct differentiation is shown on the left side of the cartoon (blue continue arrows): immature cells (CPCs, FAPs, EPCs) may directly differentiate to adipocytes when carrying the mutation. Paracrine/autocrine mechanisms for fibroadiposis are mainly shown on the right side of the cartoon (green dashed arrows): cells expressing the mutant protein may secrete molecules with autocrine and paracrine effects. The secreted factors may stimulate differentiation/activation/proliferation of neighboring cells including cells which do not express desmosomal proteins. Among these, inflammatory cells may migrate into the myocardium under the effect of chemotactic factors. EPCs: endocardial progenitor cells; EMT: epithelial to mesenchymal transition; FAPs: fibroadipocyte progenitors; CPCs: cardiac progenitor cells; CMs: cardiomyocytes; SMCs: smooth muscle cells.

**Table 1 ijms-21-06320-t001:** List of ACM causal genes.

Gene	Protein	Estimated Frequency (%)	Features	Mode of Inheritance	Refs
**Desmosome**					
PKP2	Plakophilin 2	~40	Haploinsufficiency. ACM; Carvajal syndrome; LD-ACM; Arrhythmogenic DCM	AD	[46,47,48] [45,49]
DSP	Desmoplakin	~16	biventricular ACM	AD, AR	
DSG2	Desmoglein 2	~10	Overlap with DCM; biventricular ACM	AD	[50,51]
DSC2	Desmocollin 2	~8	ACM	AD, AR	[52,53]
JUP	Junction plakoglobin	Rare	Naxos disease; ACM	AD, AR	[44,54]
**Adherens Junction**					
CTNNA3	Catenin-α3	Rare	ACM with incomplete penetrance;	AD	[65]
CDH2	Cadherin 2	Rare	No specific genotype–phenotype relationship identified	AD	[67,68]
**Cytoskeletal/Nuclear Structure**					
LMNA	Lamin A/C	Rare	DCM with arrhythmias and high risk of sudden cardiac death; LD-ACM	-	[61]
DES	Desmin	Rare	Fully penetrant; LV and RV-dominant ACM: DCM; skeletal myopathies	AD	[62,63]
FLNC	Filamin C	Rare	LD- ACM; high risk of arrhythmias and sudden death; Arrhythmogenic-DCM	-	[70,71]
TMEM43	Transmembrane protein 43	Rare	Fully penetrant; affected men more severely than women; LD-ACM	AD	[60]
TTN	Titin	Rare	Higher risk of supraventricular tachycardia and progression to heart failure; arrhythmogenic DCM	-	[64,65]
**Ion Transport**					
RYR2	Ryanodine receptor 2	Rare	Overlap with DCM	AD	[56,57]
SCN5A	Nav1.5	Rare	Prolonged QRS interval; arrhythmhenic DCM and LD-ACM	-	[66]
PLN	Phospholamban	Rare	Low prevalence; DCM,ACM; LD-ACM, biventricular ACM, arrhythogenic DCM	-	[58,72]
**Cytokine**					
TGFB3	Transforming growth factor-β3	Rare	No specific genotype–phenotype relationship identified	-	[55]

AD: autosomal dominant; AR: autosomal recessive, ACM: arrhythmogenic cardiomyopathy; LD left dominant; DCM: dilated cardiomyopathy; LV: left ventricle; RV: right ventricle; SCN5A: Sodium Voltage-Gated Channel Alpha Subunit; Nav1.5: α-subunit of the cardiac sodium channel complex.

**Table 2 ijms-21-06320-t002:** Distinctive imaging and electrocardiographic features of ARVC, LD-ACM and a-DCM

Phenotype	Chamber Involved	Histology	CMRI-LEG Distribution	ECG	Cardiac Function by Echocardiography and Cine-MRI
**ARVC**	RV	Fibroadiposis	Subepicardial enhancement. Involving the area between the anterior part of the pulmonary infundibulum, the apex, and the infero-posterior wall (so called triangle of dysplasia). Interventricular septum is not involved	Inverted T waves in right precordial leads. Epsilon wave in the right precordial leads. Non-sustained or sustained VT of LBBB morphology with superior axis	RV dilatation and reduced global function; regional RV dys- akinesias and/or RV wall aneurisms
**LD-ACM**	LV	Fibroadiposis	Subepicardial enhancement involving the inferior/lateral walls of the LV	Low voltages at limb leads; left axis deviation; negative T waves at the inferolateral leads.Ventricular extrasystoles of RBBB morphology	LV dilatation and reduced systolic function with preservation of RV global and regional function
**a-DCM**	LV or LV and RV	Interstitial Fibrosis > adiposis	Diffuse midwall enhancement of the LV; interventricular septum usually involved	Conduction system disease including sinus and/or AV nodal dysfunction, frequent atrial or ventricular arrhythmias	Mild to severe LV global dysfunction in absence of regional abnormalities; RV may or may not be involved

RV: right ventricle; LV: left ventricle; VT: ventricular tachycardia; LBBB: left bundle branch block; RBBB: right bundle branch block; AV: atrio-ventricular.

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
