# Peer review of "Established and Emerging Mechanisms in the Pathogenesis of Arrhythmogenic Cardiomyopathy: A Multifaceted Disease"

_ijms, 2020, doi:10.3390/ijms21176320_

Round 1

Reviewer 1 Report

I would liek to congratulateth eauthors on a very well written and comprehensive review of ACM.

I would only like the authors to reference statements between lines 72 and 80. I understand there might be a few studies that could not prove the theory mentioned in line 72 but it is still important to reference them.

Author Response

We thank the reviewer for the kind comments.

Point 1: I would only like the authors to reference statements between lines 72 and 80. I understand there might be a few studies that could not prove the theory mentioned in line 72 but it is still important to reference them.

Response 1: We have added the references (Ref# 39 and 40 for the theory of transdifferentiation of cardiomyocytes; and ref# 1, 20 and from 41 to 44) as requested by the reviewer. The paragraph has been rephrased as follows: “The cell origin of fibroadipocytes replacing cardiac myocytes in the context of ACM has been explained by different theories including the trans-differentiation of adult cardiomyocytes [39, 40] and the abnormal differentiation of immature cell types[1, 20, 41-44].

Reviewer 2 Report

This is a well-written review in the area of arrhythmogenic cardiomyopathy. It provided additional details on the molecular cues and in-depth discussion that other published ACM reviews did not offer. While the content is well-written, this review can be further enhanced with improved aesthetic presentation and tabular content. Suggestions are outlined below.

Major comments:

  1. The clinical hallmark of ACM is not as clearly defined in the introduction of this review. If authors choose to refer other papers for more detail description, one suggestion is to show heart section of ACM and point out the difference in this review, especially for LD-ACM. The difference or overlap of LD-ACM, aDCM and DCM can be showcased better this way.
  2. It may be helpful to include a comparison table of LD-ACM, aDCM and ARVC.
  3. Figure 2a does not contribute too much on the explanation of E-C coupling defect and arrhythmogenesis. Thus, authors should consider removing it.
  4. Figure 2b should include more players that are stated in the text, namely CX43, FKBP12.6, HRC, ANK2, TRDN and CASQ2. Authors can explain how each of these can contribute to arrhythmogenesis. Currently, both figure 2a and 2b look plain and empty.
  5. Fibroblasts are grey in color in figure 3, but it is green in the legend.
  6. Page 14, lines 527-547, the paragraph is separated into several one-sentence paragraphs. Is it a formatting issue?

Author Response

We thank the reviewer for the kind comments and the suggestions.

Point 1 and 2: The clinical hallmark of ACM is not as clearly defined in the introduction of this review. If authors choose to refer other papers for more detail description, one suggestion is to show heart section of ACM and point out the difference in this review, especially for LD-ACM. The difference or overlap of LD-ACM, aDCM and DCM can be showcased better this way. It may be helpful to include a comparison table of LD-ACM, aDCM and ARVC.

Response 1 and 2: The introduction has been shortened and a new paragraph on Genetics and clinical description of ACM phenotypes has been added. More references have been added regarding the clinical hallmarks of ACM and a more detailed description of 2 main phenotypic expressions of the disease (ARVC, LD-ACM) and aDCM has been provided. The causal genes and the related ACM phenotypes have been highlighted in table 1.Genetics and clinical overlaps between aDCM and LD-ACM, and the relative references have been discussed in paragraph 1.1 and 1.2 respectively. As the main focus of the present review is the molecular and cellular pathogenesis of ACM, the paragraph on genetics and clinical phenotype is short, but a new figure (new Fig 2) showing the genetic overlaps (panel A) and histological characteristics (panel B) of ACM and aDCM phenotypes and a summary table (Table 2) including distinctive imaging and electrocardiographic features have been included in the manuscript

Point 3: Figure 2a does not contribute too much on the explanation of E-C coupling defect and arrhythmogenesis. Thus, authors should consider removing it.

Response 3: Thank you for the suggestion and we have removed Figure 2a.

Point 4: Figure 2b should include more players that are stated in the text, namely CX43, FKBP12.6, HRC, ANK2, TRDN and CASQ2. Authors can explain how each of these can contribute to arrhythmogenesis. Currently, both figure 2a and 2b look plain and empty.

Response 4: Thank you for the suggestion. We agree with the reviewer and modified figure 2b according to reviewer’s suggestion. Please see new Figure 3.

Point 5: Fibroblasts are grey in color in figure 3, but it is green in the legend.

Response 5: Thank you. We have corrected the figure legend. Please see new Figure 3.

Point 6: Page 14, lines 527-547, the paragraph is separated into several one-sentence paragraphs. Is it a formatting issue?

Response 6: We have corrected this formatting issue. Please see section 6.

Reviewer 3 Report

This review summarizes the current knowledge about ACM mechanisms of pathogenesis. The paper is balanced and comprehensive and covers a number of studies both from the authors and from different groups. However, the same topic has already been treated in different recent reviews and the present work does not offer an extremely novel perspective.

A number of minor issues needs to be addressed:

  • Please provide references in the introduction (e.g. aDCM similarity to ACM; molecular mechanisms); moreover, all throughout the text many references are missing.
  • Figure 2 A and B are almost identical. It is not evident, at a first sight, in what the impairment consists.
  • The paragraph 3.2. “Desmosome mutations and ion channels’ function” is about Cx43 and Nav1.5. I wouldn’t define them both as ion channels. Please find a more appropriate title.
  • Please check all the apexes, both in the text and in the figures. Also, abbreviations are repeated more than once.
  • Arrhythmogenic-DCM is an undefined nosologic entity, and the overlap with ACM needs to be better defined. Also, no mention is made of the clinical and genetic overlap with CPVT or other cardiomyopathies.
  • An entire paragraph is dedicated to one PLN mutation, with many repetitions. Please shorten it.
  • Does PLN mutation provoke fibrofatty substitution? Since this characteristic is discussed for RYR2, please report what is known about PLN as well.
  • Please explain why Cx43 inhibition leads to Ca2+ homeostasis restoration (ref 64).
  • Sommariva et al. 2016 demonstrated that the blend population of stromal cells of mesenchymal origin, mainly composed of fibroblasts, do express desmosomal genes. Thus, the statement that fibroblasts do not express desmosomal genes is not correct. Please edit in figure 3 legend. Moreover, the expression of these genes in the other cell types has never been tested, to our knowledge. Please, rephrase this strong statement with milder terms (including in paragraph 5.3.).
  • The discovery of mesenchymal cells as origin of adipocytes was done almost in parallel by two groups (ref 71 and ref 74). Please acknowledge it, by correcting the sentence in lines 410-412.
  • Please discuss better the role of inflammation as a primary disease cause (Chelko et al., 2019 paper). It also has to be mentioned in the paragraph about novel therapeutic options.
  • Line 486: CSC?
  • Most of the chapter 5.2. “Paracrine factors as regulators of fibroadiposis in ACM” is based on the description of the regenerative capabilities of c-kit+ cells. However, regenerative therapy has been focusing on different types of cells, including cardiac ones and not only c-kit (which lately have been questioned). Please, either provide a comprehensive overview of regenerative cells or shorten the paragraph about c-kit cells.
  • Secreted factors are the focus of both 5.2 and part of the 5.3 paragraphs. The second part of 5.3 paragraph is about the contribution of different cell types to the pathogenesis. We suggest to reorganize the two paragraphs with clear titles.
  • The conclusion needs to be written in a more organized way and checked for English

Author Response

Point 1: Please provide references in the introduction (e.g. aDCM similarity to ACM; molecular mechanisms); moreover, all throughout the text many references are missing.

Response 1: To better define genetics and clinical manifestations of ACM we have shortened the Introduction and added a new paragraph on genetics and clinical description of ACM phenotypes (Paragraph 1, entitled: Genetics, clinical and histological hallmarks). More references have been added regarding the clinical hallmarks of ACM and a more detailed description of the 2 main phenotypic expressions of the disease (ARVC, LD-ACM) and of aDCM has been provided. The causal genes and the related ACM phenotypes have been highlighted in Table 1. Genetics and clinical overlaps between aDCM and LD-ACM, and the relative references have been discussed in paragraph 1.1 and 1.2 respectively. As the main focus of the present review is the molecular and cellular pathogenesis of ACM, the paragraph on genetics and clinical phenotype is brief, but a new figure (new Fig 2) showing the genetic overlaps (panel A) and histological characteristics (panel B) of ACM and aDCM phenotypes and a summary table (Table 2) including distinctive imaging and electrocardiographic features have been included in the manuscript. We also added the missing referenced throughout the manuscript as required by the reviewer.

Point 2: Figure 2 A and B are almost identical. It is not evident, at a first sight, in what the impairment consists.

Response 2: Figure 2A has been removed and Figure 2B has been modified according to Reviewer request. Please see new Figure 3.

Point 3: The paragraph 3.2. “Desmosome mutations and ion channels’ function” is about Cx43 and Nav1.5. I wouldn’t define them both as ion channels. Please find a more appropriate title.

Response 3: We agree with the reviewer and we have changed the title of the paragraph as follows: “Effects of desmosome mutations on the electrical properties of the cell membrane

Point 4: Please check all the apexes, both in the text and in the figures. Also, abbreviations are repeated more than once.

Response 4: Thank you for pointing this out. We have checked and updated the abbreviations.

Point 5: Arrhythmogenic-DCM is an undefined nosologic entity, and the overlap with ACM needs to be better defined. Also, no mention is made of the clinical and genetic overlap with CPVT or other cardiomyopathies.

Response 5: The overlaps between aDCM and ACM have been discussed in the new paragraph entitled: “Clinical phenotypic expressions of ACM” which also includes a new Figure (Fig.1) and a summary table (Table 2). A brief paragraph on genetic overlap between ACM and CPVT with the relative reference (#79) has been added at the end of paragraph 1.2.

Point 6: An entire paragraph is dedicated to one PLN mutation, with many repetitions. Please shorten it.

Response 6: We agree with the reviewer and shorten the paragraph. Please see section 3.3.

Point 7: Does PLN mutation provoke fibrofatty substitution? Since this characteristic is discussed for RYR2, please report what is known about PLN as well.

Response 7: Thank you for the suggestion. We have discussed this issue in section 3.3, paragraph 3 (lines 328-330).

Point 8: Please explain why Cx43 inhibition leads to Ca2+ homeostasis restoration (ref 64).

Response 8: We have provided explanation on why Cx43 inhibition leads to Ca2+ homeostasis restoration. Please see Section 3.4, paragraph 4.

Point 9: Sommariva et al. 2016 demonstrated that the blend population of stromal cells of mesenchymal origin, mainly composed of fibroblasts, do express desmosomal genes. Thus, the statement that fibroblasts do not express desmosomal genes is not correct. Please edit in figure 3 legend. Moreover, the expression of these genes in the other cell types has never been tested, to our knowledge. Please, rephrase this strong statement with milder terms (including in paragraph 5.3.).

Response 9: Cardiac Mesenchymal cells (C-MSC) described in the paper of Sommariva et al in 2016 and cardiac fibroadipocytes (FAPs) described by Lombardi et al. in the same year were isolated by different methods and by using different cell surface markers. For this reason, although an overlap is certainly a possibility (considering that FAPs are part of the mesenchymal cell pool), C-MSC and cardiac FAPs described by the 2 groups cannot be considered exactly the same cell population. This may explain the difference between the 2 papers. To isolate cardiac FAPs our group utilized fluorescence-activated cell sorting (FACS) by using a set of cell surface markers in order to select cells expressing platelet-derived growth factor receptor-α (PDGFRA) and exclude those expressing other established lineage (CD32, CD11B, CD45, Lys76, Ly−6c and Ly6c) and fibroblast markers (thymocyte differentiation antigen 1, and discoidin domain receptor 2). The population obtained by this methods was composed by 2 subsets: a fibrogenic subset expressing collagen 1 α-1 (COL1A1) and an adipogenic subset expressing CCAAT/enhancer-binding protein α, (CEBPA) transcription factor. An interesting finding in our paper was that desmoplakin was co-stained with CEBPA but not with COL1A1, indicating that desmosome protein desmoplakin was predominantly expressed in the adipogenic but not in the fibrogenic subset of cardiac FAPs. In accord with this observation, we found that desmoplakin was not expressed in cardiac adult fibroblasts and other common cardiac cell types, such as Smooth muscle cells and endothelial cells. Although we consider this finding pretty strong, as it was confirmed in both in vitro studies, as well as in vivo mapping studies using reporter mice, we acknowledge that there may be still a possibility that a certain levels of desmosome proteins may be detected in these cell types by different methodologies. Hence, to respond to reviewer concerns we have rephrased some parts of the legend of the corresponding Figure (which is Fig.4 in the revised version) and of paragraph 5.3.

Point 10: The discovery of mesenchymal cells as origin of adipocytes was done almost in parallel by two groups (ref 71 and ref 74). Please acknowledge it, by correcting the sentence in lines 410-412.

Response 10: The title of paragraph 4.3 has been changed to: “Mesenchymal cells and Cardiac fibro-adipocyte progenitors differentiation to adipocytes”. Furthermore the whole Paragraph 4.3 has been rephrased in several points to respond to the reviewer suggestions

Point 11: Please discuss better the role of inflammation as a primary disease cause (Chelko et al., 2019 paper).

Response 11: Chelko et al., 2019 paper has been added in the references (#37) and the role of inflammation as primary disease cause has been added to paragraph 5.1

Point 12: It also has to be mentioned in the paragraph about novel therapeutic options.

Response 12: A section about the therapeutic implications of Chelko et al. findings has been added to the paragraph 6 on Novel therapeutic options in ACM

Point 13: Line 486: CSC?

Response 13: We meant CPCs =cardiac progenitor cells

Point 14: Most of the chapter 5.2. “Paracrine factors as regulators of fibroadiposis in ACM” is based on the description of the regenerative capabilities of c-kit+ cells. However, regenerative therapy has been focusing on different types of cells, including cardiac ones and not only c-kit (which lately have been questioned). Please, either provide a comprehensive overview of regenerative cells or shorten the paragraph about c-kit cells.

Response 14: The role of ckit CPCs in cardiac regeneration has been shorten in paragraph 5.2

Point 15: Secreted factors are the focus of both 5.2 and part of the 5.3 paragraphs. The second part of 5.3 paragraph is about the contribution of different cell types to the pathogenesis. We suggest to reorganize the two paragraphs with clear titles.

Response 15: Paragraph 5.2 on the paracrine hypothesis has been shorten and Paragraph 5.3 has been reorganized in 2 paragraphs: “5.3. Effects of ACM causal mutations on different cell types” and “5.4. Cell types’ crosstalk in ACM”

Point 16: The conclusion needs to be written in a more organized way and checked for English

Response 16: the conclusion section has been organized and English language corrected.